# 2.4 GHz Electromagnetic Field Influences the Response of the Circadian Oscillator in the Colorectal Cancer Cell Line DLD1 to miR-34a-Mediated Regulation

**DOI:** 10.3390/ijms232113210

**Published:** 2022-10-30

**Authors:** Soňa Olejárová, Roman Moravčík, Iveta Herichová

**Affiliations:** Department of Animal Physiology and Ethology, Faculty of Natural Sciences, Comenius University Bratislava, 842 15 Bratislava, Slovakia

**Keywords:** *cry*, *per*, *bmal1*, *clock*, survivin, sirtuin1, *birc5*

## Abstract

Radiofrequency electromagnetic fields (RF-EMF) exert pleiotropic effects on biological processes including circadian rhythms. miR-34a is a small non-coding RNA whose expression is modulated by RF-EMF and has the capacity to regulate clock gene expression. However, interference between RF-EMF and miR-34a-mediated regulation of the circadian oscillator has not yet been elucidated. Therefore, the present study was designed to reveal if 24 h exposure to 2.4 GHz RF-EMF influences miR-34a-induced changes in clock gene expression, migration and proliferation in colorectal cancer cell line DLD1. The effect of up- or downregulation of miR-34a on DLD1 cells was evaluated using real-time PCR, the scratch assay test and the MTS test. Administration of miR-34a decreased the expression of *per2*, *bmal1*, *sirtuin1* and *survivin* and inhibited proliferation and migration of DLD1 cells. When miR-34a-transfected DLD1 cells were exposed to 2.4 GHz RF-EMF, an increase in *cry1* mRNA expression was observed. The inhibitory effect of miR-34a on *per2* and *survivin* was weakened and abolished, respectively. The effect of miR-34a on proliferation and migration was eliminated by RF-EMF exposure. In conclusion, RF-EMF strongly influenced regulation mediated by the tumour suppressor miR-34a on the peripheral circadian oscillator in DLD1 cells.

## 1. Introduction

The circadian system is a network of endogenous oscillators localised in the brain and peripheral tissues that facilitate proper synchronisation of the organism to environmental factors exerting a 24 h cycle [1]. The organisation of the circadian system is hierarchical, as it is composed of the central oscillator situated in the suprachiasmatic nuclei of the hypothalamus (SCN) and peripheral oscillators localised in most other tissues. Peripheral oscillators are usually under the regulatory influence of the SCN. However, there are circumstances in which peripheral oscillators can partially or completely uncouple from the master oscillator. This phenomenon arises from the different responsiveness of central and peripheral oscillators to synchronising cues. While the SCN is predominantly synchronised by the light (L) and dark (D) cycle, due to its direct connection to the retina, peripheral oscillators can be more responsive to other synchronising factors [2].

As the anticipation of cyclic changes in the environment was useful, the circadian system was evolutionarily conserved [1]. Circadian oscillator functioning relies on the expression of clock genes (*period* and *cryptochrome*), which negatively influences their own transcription. In humans, there are three homologues of the *period* gene (*pre1*, *per2* and *per3*) and two homologues of the *cryptochrome* gene (*cry1* and *cry2*). Transcription of clock genes *per* and *cry* is typically induced by heterodimers composed of transcription factors BMAL1 and CLOCK, which exert their function via the regulatory region E-box. BMAL1/CLOCK-induced transcription of clock genes promotes the accumulation of *per* and *cry* protein products in the cytoplasm. When the concentrations of PER and CRY achieve critical levels, they create heterodimers that are translocated back into the nucleus, where they inhibit the transcription of their own mRNA [3,4].

In addition to E-box-dependent regulation, there are several additional loops that influence the transcription of clock genes via RORE and DBP regulatory regions [2]. Moreover, the transcription of clock genes is also subjected to epigenetic regulation based on clock gene mRNA 3′UTR interaction with small non-coding RNAs (miRNAs). All genes of the basic feedback loop are subjected to miRNA-mediated control, although particular miRNAs usually differ for specific clock genes [5,6,7,8,9]. The administration of miR-219 or miR-132 facilitates BMAL1/CLOCK-induced expression of *per1* [10]. Similarly, *per1* expression is inhibited by miR-34a-5p [11], miR-24 and miR-29a [12]. *clock* expression is under the control of miR-182 [13], miR-17-5p [14], miR-124 [15] and miR-455-5p [16]. Expression of *bmal1* is responsive to miR-27b-3p [17], miR-142-3p, miR-494 [18,19], miR-155 [20], miR-135b [21] and miR-211 [22], and *per2* expression is inhibited by miR-24-3p and 25-3p [12,23,24], miR-30a [12,23] and miR-96 [25] and induced by miR-107 [26]. The expression of miR-34a-5p is negatively associated with *per2* mRNA levels in colorectal cancer tissue in humans [27]. The expression of the whole period family is inhibited by the miR-192/194 cluster [28]. *cry1* expression is enhanced in response to the overexpression or sponging of miR-17-5p [14], and miR-185 administration inhibits the expression of *cry1* mRNA [29]. *cry2* expression is inhibited by miR-107 [26] and miR-181d [30].

Among the cues synchronizing the circadian system are the light–dark regimen, food availability, reward, exercise and temperature [1,2,4]. In addition, the electromagnetic field has also been implicated as a possible factor with the capacity to influence circadian rhythms in the human body [31]. During evolution, living organisms were exposed only to the naturally occurring electromagnetic field of Earth [32]. However, in response to progress in science and technologies, artificially generated electromagnetic fields are growing exponentially (www.itu.int/en/ITU-D/Statistics/Documents/facts/FactsFigures2021.pdf (accessed on 26 October 2022)), and concerns about their effects on human physiology have increased, especially after the International Agency for Research on Cancer at the World Health Organization classified non-ionising radio-frequency electromagnetic fields (RF-EMF) as a Group 2B agent that is, possibly carcinogenic to humans [33]. RF-EMF include a frequency range from 30 kHz to 300 GHz and is generated mainly by cellular antennas, Wi-Fi access points and Bluetooth devices [34]. The intensity of radiofrequencies (RF) increased 2.3 times from 2017 to 2020, with Wi-Fi as the most important contributor [35].

Except for indirect heat effects of electromagnetic fields on living organisms, numerous direct non-thermal interactions have been reported recently. RF-EMF can induce electron and/or ion vibrations and consequently interactions among molecules that would not occur otherwise. It was shown that RF-EMF influence polarisation of charged particles and generation of electric dipoles. Moreover, effects of RF-EMF on potentials of cellular membranes and reactive oxygen species levels have been reported [36]. Considering constantly growing intensities of electromagnetic fields used in urban areas, non-thermal effects of RF-EMF definitely deserve attention.

There is little data on the effect of Wi-Fi on clock gene expression. The effect of RF-EMF on the circadian system has been investigated with respect to reproductive functions, where the expression of *clock*, *bmal1* and *rorα* mRNA is strongly suppressed in Leydig cells of male mice exposed to1.8 GHz RF for several hours [35]. Similarly, the expression of clock genes *per*, *clc* and *cyc* is downregulated, while *cry* expression is strongly induced by3.5 GHz RF-EMF in *Drosophila* [37].

Clock genes can influence the cell cycle [38,39], and deregulation of the circadian system has been associated with cancer progression [40,41]. A regulatory relationship between clock gene expression and cell cycle has been demonstrated in many types of cancer [42].

Downregulation of *per2* was associated with tumour progression in non-small cell lung cancer [43]. Similarly, over-expression of *per2* inhibited growth of lung carcinoma and murine mammary carcinoma cell lines LLC and EMT6, respectively [44]. Over-expression of *per2* inhibited growth and promoted apoptosis of K562 leukemia cells [45]. On the other hand, *bmal1* has been shown to induce metastasis, migration and invasion in ZR-75-30 and MDA-MB-231 lines of breast cancer [46]. Similarly, the *clock* gene has been reported to induce proliferation of breast cancer cell lines MCF-7, T47D and MDA-MB-231 [47]. In the HL-60 promyeloblast cell line, *bmal1* downregulation induced apoptosis and inhibited proliferation [48]. The opposite results were observed in tongue squamous carcinoma cell lines SCC9, SCC25, and CAL27 [49] and nasopharyngeal carcinoma cell line [50], where over-expression of *bmal1* inhibited cell proliferation. Upregulation of *bmal1* has been shown to inhibit proliferation U87MG glioblastoma cells [51] and *per2* inhibited proliferation and invasion ability in glioma stem cells U87 and U251 [52]. The effect of *bmal1* downregulation on growth of colorectal cancer (CRC) cell lines (HCT116 and SW481) and the metastatic CRC line SW620 was dependent on the functional p53 pathway and AKT/mTOR activity. Proliferation was increased in HCT116 and SW620 cells while inhibition of cell growth was observed in SW480 cells after *bmal1* knockdown [53].A functional relationship between β-catenin and *per2* stability in CRC cell lines HCT116 and SW480 implicates a role of the circadian oscillator in β-catenin- mediated effects [54].

Among cancer-related diseases, colorectal cancer has been identified as the third most commonly diagnosed malignancy, with the second highest mortality rate after lung cancer [55]. Moreover, during the Covid-19 pandemic, cancer screening and routine diagnostics dropped. The estimated increase in deaths because of colorectal cancer (15–16%) was highest among the four types of cancer (colorectal, breast, lung, and oesophageal) that were included in the analysis [56].

Studies focused on clock genes expression in colorectal cancer tissue have revealed that their expression is frequently down- or deregulated in cancer tissue [57]. Previously, we observed downregulation of *cry2* and *per2* expression in tumour tissue in comparison to adjacent tissue [27], and higher expression of *cry1* in right-sided tumours but not in left-sided tumours compared to surrounding tissue [58]. A negative association between *per2* expression and tumour staging has also been described [59]. Similarly, the amplitude and mesor of *bmal1*, *per1*, *per2* and *rev-erba* decreased in cancer tissue compared to healthy colon in a mice model with induced colorectal tumours [60].

Considering that clock genes are involved in the regulation of cancer progression and that their expression is modulated by miRNAs, there is quite a complex regulatory network influencing processes in cancer tissue during the 24 h cycle. Our previous research implicated the role of miR-34a in *per2* regulation in patients with higher TNM stages [61]. Accordingly, miR-34a has been shown to have strong potential in colorectal cancer treatment [62,63] and its prognosis [64,65,66]. Recently, concern about the effect of RF-EMF on clock gene expression has risen, as the use of RF-EMF have significantly increased during recent years and colorectal cancer treatment has been weakened by overwhelmed medical capacities due to the Covid-19 pandemic. Therefore, this study was designed to determine whether RF-EMF influence miR-34a-mediated regulation of clock gene expression in the colorectal cell line DLD1 and whether it can be associated with cancer cell line growth.

## 2. Results

qPCR: Two-way ANOVA confirmed that the administration of miR-34a mimics (48 h) resulted in a significant increase in the miR-34a-5p level in transfected cells (m-m: F _[2, 15]_ = 31.42, *p* < 0.0001, Figure 1A; pre-m: F _[2, 17]_ = 83.92, *p* < 0.0001, Figure 1B). A post hoc test revealed a significant increase in miR-34a levels in cells influenced by the mimic or the mimic together with the inhibitor compared to the corresponding control (Tukey’s post hoc test; Figure 1A,B). We did not observe a significant effect of RF-EMF administration (24 h) on the intracellular levels of miR-34a (m-m: F _[1, 15]_ = 0.61, *p* = 0.4474; pre-m: F _[1, 15]_ = 1.83, *p* = 0.1963), indicating that RF-EMF did not influence transfection and/or levels of miR-34a in transfected cells. Accordingly, the interaction between the investigated factors was not confirmed (two-way ANOVA).

A two-way ANOVA confirmed the interaction between the influences of transfected oligos and RF-EMF (m-m: F _[2, 17]_ = 3.759, *p* = 0.067; pre-m: F _[2, 17]_ = 8.143, *p* < 0.01) and the effect of miR-34a administration (m-m: F _[2, 17]_ = 3.759, *p* = 0.045, Figure 2A; pre-m: F _[2, 17]_ = 15.96, *p* < 0.0001, Figure 2B) on *cry1* mRNA levels. Accordingly, *cry1* mRNA expression was significantly induced by miR-34a administration only in cells exposed to RF-EMF, while no effect of mi-34a on *cry1* mRNA expression was observed under control conditions (Tukey’s post hoc test; Figure 2A,B). We also revealed an effect of RF-EMF on *cry1* mRNA (m-m: F _[1, 17]_ = 3.981, *p* = 0.062; pre-m: F _[1, 17]_ = 12.03, *p* < 0.01; followed by Šídák’s multiple comparisons test, *p* < 0.05 and *p* < 0.001, respectively, Figure 2A,B). *cry1* mRNA levels were suppressed by RF-EMF when miR-34a was not administered in cells transfected with mature oligos (Šídák’s multiple comparisons test; Figure 2A).

miR-34a administration significantly influenced *cry2* mRNA expression (pre-m: F _[2, 17]_= 5.116, *p* < 0.05), which was reflected by a trend and an increase in *cry2* mRNA levels in cells transfected with miR-34a or miR-34a + inhibitor relative to the negative control (Tukey’s post hoc test, *p* = 0.087 and *p* < 0.05, respectively, Figure 3B) under Wi-Fi-free conditions. Two-way ANOVA also proved the effect of RF-EMF on *cry2* mRNA expression (m-m: F _[1, 17]_ = 8.519, *p* < 0.01, Figure 3A; pre-m: F _[1, 17]_ = 13.99, *p* < 0.01, Figure 3B), and the post hoc test revealed a significant increase in *cry2* mRNA expression in groups influenced by the mimic together with the inhibitor of miR-34a with a significant decrease under RF-EMF conditions compared to the Wi-Fi-free environment (Šídák’s multiple comparisons test, Figure 3A,B). Two-way ANOVA did not show interference between the investigated factors (miR-34a and RF-EMF).

As interference between factors was not proven, the results were also analysed using simple ANOVA for oligo type and treatment. ANOVA was used when three groups were compared (effect of oligos), and an unpaired *t*-test was employed to compare the treatments. In this way, we revealed an RF-EMF-associated decrease in *cry2* mRNA expression in groups mNC, preNC and pre-m compared to Wi-Fi-free conditions (*t*-test, *p* < 0.05). Interestingly, despite this, significantly higher expression of *cry2* mRNA was detected in cells transfected with mir-34a compared to the corresponding control under RF-EMF conditions (pre-m: One-way ANOVA: F _[2, 9]_ = 5.598, *p* = 0.026, *p* < 0.05, Figure 3B).

Two-way ANOVA clearly indicated a significant influence of miR-34a administration (m-m: F _[2, 17]_ = 28.14, *p* < 0.0001, Figure 4A; pre-m: F _[2, 17]_ = 20.77; *p* < 0.0001, Figure 4B) and the effect of RF-EMF (m-m: F _[1, 17]_ = 5.356, *p* < 0.05, Figure 4A; pre-m: F _[1, 17]_ = 6.039, *p* < 0.05, Figure 4B) on *per2* mRNA expression. The interaction of the investigated factors, however, was not indicated.

The post hoc test demonstrated that miR-34a strongly inhibits *per2* expression compared to the negative control under Wi-Fi-free conditions. Moreover, the miR-34a inhibitor reversed this effect and caused an increase in *per2* expression compared to the control (Tukey’s post hoc test; *p* < 0.05; Figure 4A,B). Exposure of DLD1 cells to RF-EMF inhibited miR-34a-mediated effects on *per2* mRNA expression (Tukey’s post hoc test; Figure 4A,B). We also observed a significant effect of RF-EMF on *per2* expression in groups that were treated with the mimic and inhibitor (m-m + i) together, where a significant decrease in *per2* mRNA expression was detected as a result of RF-EMF exposure (Šídák’s multiple comparisons test; Figure 4A).

The statistical analysis implicated a significant effect of transfected oligos on *clock* mRNA expression (m-m: F _[2, 18]_ = 4.58, *p* < 0.05, Figure 5A; pre-m: F _[2, 18]_ = 8.04, *p* < 0.01, Figure 5B), and the post hoc analysis confirmed a difference between cells transfected with the mimic + inhibitor compared to the control (Tukey’s post hoc test, *p* < 0.05, Figure 5B). Two-way ANOVA did not reveal an effect of RF-EMF exposure on *clock* mRNA expression or an interaction between the tested factors.

Two-way ANOVA revealed a significant effect of miR-34a on *bmal1* mRNA expression (m-m: F _[2, 17]_ = 7.879, *p* < 0.01, Figure 6A; pre-m: F _[2, 17]_ = 10.31, *p* < 0.01, Figure 6B), and a post hoc test confirmed a significant decrease in *bmal1* levels in cells transfected with miR-34a compared to control and/or cells transfected with the mimic + inhibitor under control conditions (Tukey’s post hoc test, Figure 6A,B). RF-EMF exposure also influenced *bmal1* levels in DLD1 cells (m-m: F _[1, 17]_ = 6.023; *p* < 0.05), and a post hoc test confirmed a significantly higher level of *bmal1* mRNA expression in cells transfected with the miR-34a mimic alone cultured in the RF-EMF environment than expression measured in the corresponding group cultured under Wi-Fi-free conditions (Šídák’s multiple comparisons test, Figure 6A). This increase may be related to the diminishing of inhibitory influence of miR-34a observed under the control conditions. Interference between tested factors was not significant, although it was close (m-m: F _[2, 17]_ = 3.389; *p* = 0.06).

Two-way ANOVA confirmed the effect of transfected oligos (m-m: F _[2, 17]_ = 9.988, *p* < 0.01, Figure 1A; pre-m: F _[2, 17]_ = 11.17, *p* < 0.001) on *sirt1* mRNA expression regardless of the RF-EMF factor. Post hoc tests revealed a significant decrease in *sirt1* mRNA expression in miR-34a transfected cells compared to those also transfected with the inhibitor under control and RF-EMF conditions (Tukey’s and Šídák’s multiple comparisons test, respectively; Figure 7A,B). Statistical analysis did not indicate an interaction of the tested factors.

Expression of *survivin* was inhibited by miR-34a transfection (m-m: F _[2, 17]_ = 8.753, *p* < 0.01, Figure 8A; pre-m: F _[2, 17]_ = 5.412, *p* < 0.05, Figure 8B) and Tukey’s post hoc test confirmed a significant decrease in *survivin* expression in cells transfected with miR-34a compared to the control or compared to the mimic + inhibitor (Šídák’s multiple comparisons test, Figure 8A,B) in Wi-Fi-free conditions. Two-way ANOVA was not significant for RF-EMF (m-m: F _[1, 17]_ = 3.259; *p* = 0.08); however, the decrease in *survivin* expression after miR-34a administration observed in the Wi-Fi-free environment, according to Tukey’s post hoc, diminished in cells exposed to RF-EMF (Figure 8A,B).

Scratch assay: A two-way ANOVA indicated a significant inhibitory effect of miR-34a (48 h) on wound closure (pre-m: F _[2, 18]_ = 8.232; *p* < 0.01, Figure 9B) and Tukey’s post hoc test confirmed significantly slower wound closure in cells transfected with miR-34a compared to cells transfected with the mimic together with the inhibitor or to the negative control under Wi-Fi-free conditions (Figure 9B). This effect was not observed in cells transfected with miR-34a exposed to RF-EMF for 24 h (Tukey’s post hoc test, Figure 9B).

MTS assay: The results indicate a significant effect of RF-EMF exposure (48 h) on proliferation (pre-m: F _[1, 83]_ = 7.215, *p* < 0.01, Figure 10B) and the interaction between the RF-EMF and transfected oligos (pre-m: F _[2, 83]_ = 6.330, *p* < 0.01). Post hoc analysis confirmed that miR-34a (48 h) reduced the proliferation of DLD1 cells compared to the corresponding control under Wi-Fi-free conditions (Tukey’s post hoc test; *p* < 0.01; Figure 10B). Proliferation showed a decreasing trend (*p* = 0.08) in miR-34a-treated cells compared to cells transfected with the mimic together with the inhibitor under control conditions (Tukey’s post hoc test; *p* < 0.01; Figure 10B). Exposure of cells to RF-EMF caused a diminishing of miR-34a-induced inhibition in cell proliferation under Wi-Fi-free conditions (Šídák’s multiple comparisons test, Figure 10B). Instead, we observed an increase in proliferation in cells exposed to miR-34a together with the inhibitor compared to the corresponding control under RF-EMF conditions (Šídák’s multiple comparisons test, Figure 10A,B).

In most cases, the administration of the mature dominant strand and precursor form of miR-34a-5p caused a similar effect. In some cases, however, more pronounced results were observed when pre-miR-34a was administered instead of miR-34a-5p. This finding is addressed in the discussion in more detail. However, as the administration of precursor and mature forms of miR-34a increased the intracellular level of miR-34a-5p identically (Figure 1A,B), we consider both treatments to be equally informative.

## 3. Discussion

Recent data indicate that 2.4 GHz RF-EMF influence the effect of miR-34a on the transcription of clock genes and *survivin* (*birc5*), an inhibitor of apoptosis. Under control conditions, miR-34a administration caused a significant decrease in *per2*, *bmal1*, *sirt1* and *survivin* mRNA expression. However, the exposure of DLD1 cells to RF-EMF resulted in weakening or diminishing of the miR-34a effect on *per2* and *survivin*, respectively. Moreover, RF-EMF administration was accompanied by a significant increase in *cry1* expression in cells treated with miR-34a. The effect of RF-EMF was also reflected at the level of cell migration and viability, as miR-34a significantly decreased wound closure and metabolism intensity in cell culture under control conditions, while this effect was not observed when cells were exposed to RF-EMF.

A decrease in *per2* expression after miR-34a has been implicated previously by in silico analysis [67], cross-linking immunoprecipitation sequencing (GSE161238, GSE161239), a negative correlation between *per2* expression and miR-34a was observed in colorectal cancer tissue [27] and recent results clearly confirmed that miR-34a inhibits the expression of *per2* under in vitro conditions.

A decrease in *bmal1* expression observed after miR-34a administration is in agreement with the in silico analysis performed by miRWalk [68]. However, we did not observe a decrease in *clock* gene expression in miR-34a-treated cells, which was implicated by in silico analysis [67] and observed previously in human oesophageal epithelium [69], which can be attributed to different cell types and/or methodological differences in the experimental setups. *cry1* is a predicted target of miR-34a according to miRWalk [68], and *cry2* has been predicted to be a target of miR-34a by TargetScan [67]. However, in a recent study, we observed an increase in *cry* genes expression induced by miR-34a administration, which implicates other than 3′UTR-mediated regulation.

It was shown that miRNAs (typically inhibitors) can also induce gene expression. In particular, one of the pioneer studies showed that miR-373 induces the gene expression of E-cadherin via its complementary region in the promoter sequence [70]. A similar mechanism of gene expression induction was also shown for other genes, and it was revealed that miRNAs localised in the nucleus can, in cooperation with Argonaute proteins (AGO), interact with gene promoters and either activate or inhibit transcription [71]. In silico analysis indicated the presence of a miRNA-responsive element with 75% homology with a seed sequence of miR-34a-5p in the promoter region of *cry1* and no responsive element in the UTR region [67,72]. Thus, upregulation of *cry1* expression via AGO-mediated interaction is a possible mechanism for how miR-34a could induce *cry1* expression under constant conditions. Interestingly, this upregulation was potentiated under RF-EMF conditions.

In accordance with an in silico analysis [67], we observed an inhibition of *sirt1* expression after mir-34a administration. Inhibition of *sirt1* expression had been previously shown in colorectal cancer cell lines HCT-116 [73,74], DLD1 [75], SW480 [76] and a kidney epithelial cell line [77], and our data are in accordance with these observations.

The inhibition of *survivin* expression by miR-34a was also predicted by in silico analysis [67]. The level of *survivin* mRNA expression was significantly decreased after 72 h of incubation with miR-34a in the triple-negative breast cancer cell lines MDA-MB-231 and SUM-159 [78]. A similar regulatory relationship between *survivin* and miR-34a was demonstrated in several gastric cancer cell lines [79,80,81]. The p53-dependent inhibition of *survivin* expression induced by mir-34a administration was demonstrated in colorectal cancer cell line HCT116 [82]. We observed pronounced downregulation of *survivin* expression after miR-34a administration in DLD1 cells, which is in accordance with previously reported results.

Under recent experimental conditions, RF-EMF treatment was associated with significant changes in the miR-34a-mediated regulation of five out of seven investigated target genes. In cells exposed to RF-EMF, the downregulation of *per2* and *survivin* expression caused by miR-34a was weakened or eliminated, respectively, and expression of *cry1* was stimulated more by miR-34a in RF-EMF-treated cells compared to the control. In addition to these major effects, two-way ANOVA also indicated a significant influence of RF-EMF on the *cry2* and *bmal1* mRNA levels.

Recently, it is not known how 2.4 GHz RF-EMF cause these effects. As mentioned above, clock genes are functionally interconnected via feedback loops and influence the expression and/or functioning of each other [83,84,85,86,87]. If one of them is a functional sensor of RF-EMF, this sensor can mediate the influence of RF-EMF on the expression of other clock genes.

In this respect, the most likely candidates are cryptochromes. Cryptochromes are evolutionary descendants of light-activated DNA repair enzymes that in mammals lost the capacity to bind DNA and repair UV-induced DNA photoproducts. Instead, the role of mammalian CRY1 and CRY2 proteins in the circadian loop influences gene expression via the E-box, which emerged during evolution. The functioning of *cry* genes in the generation of the circadian pattern of locomotor activity [88] and their capacity to influence synchronisation by light has been proven in double mutant mice *cry1^-/-^*and *cry2^-/-^* [89,90].

In several insect and vertebrate species, the role of magnetoreception has been attributed to CRY proteins [91,92], and the role of the CRY protein as an effector molecule transmitting magnetosensing to the circadian system has been proven, at least in *Drosophila* [93,94]. However, this function is believed to be based on photo-excitation of flavin adenine dinucleotide (FAD) facilitated by the Trp-triad located in the CRY molecule, and the structure of the FAD-binding domain in mammalian CRY proteins does not allow sufficient FAD binding. Therefore, the role of CRY proteins in the mediation of electromagnetic fields to the circadian system is not expected to be executed in this way in humans [91].

Despite the weak flavin binding affinity for mammalian CRYs, the role of human CRY2 (hCRY2) in light-dependent magnetoreception was demonstrated, as in *cry*-deficient *Drosophila*, responsiveness to the magnetic field was rescued by the insertion of human *cry2* into the fly genome [95]. In a similar experiment with the use of humanised *Drosophila*, the human *cry1* (hCRY1) insert [96] restored the behavioural response of *cry*-deficient flies to a weak pulsed electromagnetic field (PEMF) with a frequency of 10 Hz. Moreover, the PEMF strongly induced reactive oxygen species (ROS) generation and influenced the transcriptome [97]. ROS generation in response to RF-EMF was also reported in human HEK293 cells where the 1.8 GHz field strongly modified the expression of antioxidative and oxidative enzymes [98]. The response of CRY1 to a magnetic field up to 1.32 mT under in vitro conditions has also been demonstrated [99].

Therefore, there is still uncertainty about the role of FAD activation in the CRY-mediated response to the magnetic and/or electromagnetic field, and several alternative hypotheses have been proposed. One of them, based on a mutational study of CRY protein, showed that extremely low frequencies (3–50 Hz) modify the free running period in *Drosophila,* even in the case where only the C-terminus of the CRY protein, which does not contain the Trp-triad and FAD-binding pocket, is preserved. This observation suggests a mechanism of electromagnetic field-induced effects on the circadian oscillator other than that based on radical pairs theory and FAD activation [94].

Another open question is whether sensing of experimentally tested magnetic and electromagnetic fields generally share a common mechanism and, if so, how broad a range of intensities and frequencies are covered by this pathway. The argument for a unique mechanism relies on the fact that a moving charge produces a magnetic field and therefore electric and magnetic fields usually go together. Functional similarity between magnetosensing and sensing of RF-EMF is implicated by studies evidencing that RF-EMF can influence pathway(s) involved in magnetoreception, as RF-EMF of anthropogenic origin, even with very low magnitude, disturb magnetoreceptor-based orientation [100]. The present results implicate the role of the CRY1 protein in RF-EMF-mediated effects under the specific conditions used in our study.

Although the deregulation of clock genes under RF-EMF conditions could be explained by the sensory role of *cry* genes, the mechanism by which *survivin* expression can respond to RF-EMF is completely unknown. A functional E-box was not revealed in the *survivin* sequence, and *survivin* mRNA did not show rhythmic expression [101]. However, previously, *survivin* expression was shown to have a negative correlation with the clock gene *per2* in patients with colorectal cancer [102]; thus, it is possible that *survivin* expression is functionally related to the circadian oscillator indirectly, at least in some specific biological context.

Interestingly, RF-EMF effects of miR-34a are gene-specific, which implies the possible involvement of modified interactions of miRNA and mRNA. This hypothesis is supported by the observation that the electromagnetic field can influence hybridisation [103]; however, in the context of RF-EMF frequency and the type of nucleic acid used, this assumption needs to be evaluated.

Taken together, it is likely that more than one factor contributes to the gene-specific response of RF-EMF-modulated effects of mR-34a.

Previously, the expression of miRNA was shown to be influenced by a wide spectrum of electromagnetic field wavelengths. A 50 Hz electromagnetic field exposure lasting for 60 days caused a decrease in miRNA levels in the brain and circulation of rats in a sex- and age-dependent manner [104]. Long-time exposure to a 900 MHz RF-EMF caused a decrease in miR-107 expression in the brain [105], and long-time exposure to a 2.4 GHz RF-EMF caused changes in miR-106b, miR-107 and miR-181 expression in rat brain [106,107].

Under recent experimental conditions, we did not observe the effect of RF-EMF directly on the miR-34a level in cells, but the effects exerted by this miRNA changed. As many target genes of miR-34a, including *sirt1* [108] and *survivin* [109], have been shown to promote cancer progression and/or were associated with poor patient prognosis, miR-34a is frequently referred to as a tumour suppressor [62]. However, diverse roles of newly recognised target genes of miR-34a with respect to tumorigenesis were reported.

Overexpression of *cry1* stimulated proliferation, colony formation and cell migration in the HCT116 colorectal cell line and silencing of *cry1* caused the inhibition of SW480 cell proliferation. Under in vivo conditions, *cry1* overexpression promoted colorectal cancer growth [110]. High levels of *cry1* were associated with the development of metachronous metastasis [111] and worse survival in patients with colorectal cancer [27,110,111,112]. Upregulation of *cry1* inhibited apoptosis in colorectal cell lines HT29 and SW480, increased proliferation in HCT116, HT29 and SW480, and decreased responsiveness to 5-fluorouracil (5-FU) in HCT116 and SW480 cells [112]. However, downregulation of *cry1* caused an increase in osteosarcoma cell proliferation and migration and stimulated tumour growth in nude mice under in vivo conditions [113]. Despite some contradictory reports in CRC models, oncogenic features are usually attributed to *cry1* in respect to CRC and gastric cancer progression [114].

Similarly, as in the case of *cry1*, oncogenic effects were predominantly referred with respect to the *cry2* gene, as worse survival was associated with high *cry2* expression in colorectal cancer tissue [27,112,115]. Downregulation of *cry2* expression in DLD1 and SW480 cells showed pro-apoptotic effects of oxaliplatin (OXA) [115]. *cry2* overexpression in colorectal cell lines CaCo2 HT29 and SW480 inhibited apoptosis, increased proliferation in HT29 and SW480 cells and decreased the response to 5-fluorouracil (5-FU) in HCT116 and SW480 CRC cells [112].

On the other hand, *bmal1* exerts tumour-suppressive effects. *bmal1* silencing increased C26 cell proliferation in mice bearing C26 cell-derived tumours. Similarly, *bmal1* downregulation caused a decrease in Etoposid-induced apoptosis and DNA damage induced by cisplatin administration [116]. *bmal1* silencing also caused a decreased rate of proliferation in the SW480 CRC cell line, an increase in glycolytic activity, and a modified response to the glucose transport inhibitor (VZB117) and OXA in SW480 and SW620 CRC cell lines [117]. Moreover, a high level of *bmal1* was associated with better survival compared to low *bmal1* expression in CRC patients [111].

Previously, the tumour-suppressive effects of *per2* were referred to [118], as *per2*-deficient mice were more susceptible to tumour induction by γ radiation compared to wild-type mice. *per2* mutant mice also showed a decrease in p53 and an increase in c-Myc and cytochrome *c* response to γ radiation compared to the control. Therefore, *per2* can influence the DNA damage response [119]. In accordance with a previous observation, *per2* downregulation was associated with attenuated apoptosis and a delay in CHK2 response to double-stranded DNA breaks induced by doxorubicin in the HCT116 cell line [120]. *per2* inactivation increased cell proliferation, β-catenin and D cyclin levels in HCT116 and SW480 cell lines and increased small intestinal and colon polyp numbers in Apc mutant mice [121]. Overexpression of *per2* sensitised cancer cell lines Panc1 and Aspc1 to cisplatin, inhibited cell proliferation and induced apoptosis [85]. The tumour- suppressive effect of *per2* was demonstrated in oral squamous cell carcinoma (OSCC), as *per2* silencing inhibited autophagy, apoptosis and increased the proliferation rate [122,123]. Moreover, *per2* silencing significantly decreased the expression of p53, p16 and p21 mRNA and increased mRNA expression of cyclin A2, B1 and D1, and CDK4, CDK6 and E2F1 in OSCC Tca8113 cells. These results imply that *per2* plays an important role in cell cycle progression [123]. Although better survival was found in CRC patients with high compared to low *per2* expression [111], in this aspect complete agreement has not yet been achieved, as the association of *per2* with survival has not been consistently observed [27,124].

In respect to the known capacity of miR-34a to attenuate tumour growth [66], its influence on the circadian oscillator differs from that usually reported, as clock genes that are inhibited by miR-34 administration (*per2* and *bmal1*) are more often mentioned in the context of tumour suppression than the opposite. However, clock genes are not major effectors in miR-34a-mediated effects on cancer progression, and we suppose that effects mediated by *per2* and *bmal1* are under control conditions overwhelmed by the effect of other target genes of miR-34a, whose oncogenic capacity (e.g., *sirt1* and *survivin*) is higher than the tumour-suppressive capacity of the above-mentioned clock genes.

However, transfection of miR-34a increased the expression of *cry1*. This observation is in accordance with a previous study referring to the positive correlation between miRNAs and their target clock genes [125], although the mechanism of this regulation has not been entirely elucidated. *Cry1* is frequently referred to as a tumour promoter in colorectal cancer [114]. Under the conditions of a recent study, miR-34a induced *cry1* expression only when cells were exposed to RF-EMF. Interestingly, the inhibitory effect of miR-34a on *survivin* expression diminished when the cells were exposed to RF-EMF, implicating the gene-specific effect of RF-EMF on miR-34a-induced regulation. Although the mechanism of this regulation is not known, it seems that RF-EMF have the capacity to switch the role of miR-34a in cancer progression from typical tumour suppressor to neutral or slightly oncogenic (at least in respect to the studied target genes).

This observation is in line with the RF-EMF-induced elimination of the decrease in cell proliferation and migration observed after miR-34a administration under control conditions.

Generally, miR-34a is known to inhibit cell proliferation and migration [82,126,127], which was confirmed in our study. In contrast, both *cry1* and *survivin* have previously been shown to possess the capacity to exert opposite effects [110,112,128,129]. Comprehensive meta-analysis confirmed that RF-EMF predominantly influence the growth of human and faster-growing cells (45% and 47% of the included studies, respectively), and the observed effects are cell type- and dose-dependent under in vitro [130] and in vivo [131,132,133] conditions. However, in HCT-116 and DLD1 cells, RF-EMF from 900–2100 MHz applied up to 4 h did not induce changes in proliferation, as assessed by the MTT test [134]. These results are in line with our study, as we also did not observe a significant effect of RF-EMF on cell proliferation when miR-34a was not administered. However, human colonic adenocarcinoma cells (Caco-2) were responsive to2.5GHz RF-EMF, which caused a decrease in cell proliferation [135]. Therefore, even within clusters of colorectal cell lines, the genetic background and/or transcriptome state of the cells can be important in the manifestation of RF-EMF-mediated effects.

Although in most cases the effect of the mature form of the miR-34a dominant strand and pre-miRNA showed very similar effects on gene expression, in some cases, a more significant effect was observed (usually, administration of pre-miRNA was more potent). We suspect that this effect can be caused by different rates of mature and premature miRNA incorporation into the RISC complex (under normal conditions, pre-miRNA is bound by DICER and incorporated into the RISC complex) and/or turnover half-time of pre-miRNA and mature miRNA [136,137]. We previously observed a different pattern in miR-34a-5p and its premature form, implicating a non-uniform rate of their processing [138]. An alternative explanation for differences in the output of miRNA vs. pre-miRNA administration can be related to the involvement of the loop structure in the interference reaction. This feature of miRNA-mediated regulation was observed previously when it was revealed [139] that pre-miRNA can be more effective in the treatment of human gastric cancer cells compared to the miR-34a mimic, as in some cases loop sequences can contribute to the efficiency of gene targeting [140]. Knowledge about the possible role of loop sequences in target recognition is also being incorporated into new miRNA-based therapeutics [141].

As studies aimed at the effects of RF-EMF often bring inconclusive results, observed effects frequently do not show dose and/or time dependence or the response curve has several peaks [31,130], it is difficult to make firm conclusions about the impact of RF-EMF on the state of a particular biological system. Because of this broadly observed obstacle, the attributive hypothesis has been postulated. Attributive theory is based on the existence of pre-existing conditions whose presence allows to reveal the RF-EMF-mediated effect [142]. Our results are in complete agreement with this hypothesis, as under the conditions of a recent study, RF-EMF exerted most of their effects only in cases when another treatment was administered concurrently.

## 4. Materials and Methods

The human colon adenocarcinoma cell line DLD1 (CCL-221) was obtained from the American Type Culture Collection (ATCC, Manassas, VA, USA). DLD1 cells were cultured in RPMI 1640 medium (Thermo Fisher Scientific,Waltham, MA, USA) supplemented with 10% foetal bovine serum (FBS, Biosera, Nuaille, France), penicillin (50 U/mL) (Gibco, St. Louis, MO, USA), streptomycin (50μL/mL) (Gibco, USA), and ampicillin (50 μg/mL) (Oasis-lab, San Francisco, CA, USA) in biological Celculture^®^ Incubator CCL-050B-8 (Esco medical, Egå, Denmark) with a humidified atmosphere containing 5% CO_2_ at 37 °C. The 24-well plates were covered with 1% sterile gelatin.

To test the effect of RF-EMF on gene expression, DLD1 cells were exposed to a pulsed electromagnetic field, generated by a D-Link GO-RT-N150 Wi-Fi router (D-Link, Taipei, Taiwan), during 24 h. The radiofrequency range was 2426 to 2448 MHz (Wi-Fi channel 6), pulse length was 2.76 ms, pulse frequency was 9.7 Hz, pulse risetime was 0.06–0.08 µs, and pulse falltime was 0.067–0.107 µs. 

Radiofrequency field power flux density at the level of the cell layer was 1 W/m^2^ (19 V/m) peak, 0.12 W/m^2^ (6.6 V/m) RMS. Plates with sham-exposed cells were during this time covered by radiofrequency protective foil YSHIELD HNV100 (YSHIELD GmbH & Co. KG, Ruhstorf an der Rott, Germany). Other conditions were identical for both groups: powerline frequency electric field (50 Hz) at the level of cell layer was below 1 V/m, powerline frequency magnetic field at the level of cell layer was 0.3 µT in both control and experimental groups.

To manipulate intracellular miR-34a levels in the DLD1 cells, Lipofectamine^®^RNAiMAX Reagent (Thermo Fisher Scientific, USA) and Opti-MEM medium (Thermofisher, USA) were used according to the manufacturer’s instructions. Cells were transfected with a mimic of miR-34a mature dominant strand miR-34a-5p (m-m), either with a precursor of miR-34a mimic (pre-m) or with a miR-34a-inhibitor and m-m (m-m + i, Catalogue #4464066, #AM17100, #4464066, respectively, Life Technologies, Carlsbad, CA, USA) at a concentration of 100 nM. Control cells were transfected with the corresponding negative controls (mNC #4464058, preNC #AM17110, respectively, Life Technologies, USA). The effect of RF-EMF was tested in the following groups: mimic of pre-miR-34a, NC pre-miR-34a, mimic miR-34a-5p, NC miR-34a-5p and mimic + inhibitor miR-34a-5p. Oligos were added to cell culture immediately after cell trypsinisation to facilitate the entrance of nucleotides into cells, and cells were seeded into 24- or 96-well plates in concentrations of 1 mil. or 0.25 mil. cells per well, respectively (Appendix A).

Scratch assay: The effect of 24 h exposure to RF-EMF on cell migration was tested when the cell culture reached a confluence of 80–90% (24 h after transfection). The monolayers were scratched using a 10μLsterile tip. Pictures were taken immediately after wound generation and 24 h after the scratch assay with an inverted fluorescence microscope NIB-100F (Nanjing Jiangnan Novel Optics Co.,Ltd., Nanjing, China) and BEL Capture 3.2 software (BEL Engineering s.r.l., Monza, Italy). Wound closing was analysed using TScratch 1.0 software (ETH, Zurich, Switzerland, [130]. Consequently, the cells were harvested for gene expression analysis, as described below. 

MTS assay: The effect of 24 h RF-EMF exposure on the metabolism of viable proliferating cells was determined by an MTS test (CellTiter 96 AQ_ueous_ Cell Proliferation Assay, Promega, Madison, WI, USA) employing the modified tetrazolium compound (MTS), whose conversion reflects the activity of mitochondrial dehydrogenase producing NADPH or NADH in metabolically active cells. The MTS test was performed according to the manufacturer’s instructions 3 h after MTS administration, during which the cells were exposed to RF-EMF or incubated under control conditions at 37 °C and 5% CO_2_. The reaction was terminated by adding 10% sodium dodecyl sulphate, and absorbance was measured at 490 nm using a UV spectrophotometer (Epoch, Agilent Technologies, Inc., Santa Clara, CA, USA). 

qPCR: The extraction of mRNA and miRNA from DLD1 cells was performed after 24 h of RF-EMF treatment using RNAzol according to the manufacturer’s instructions (MRC, Washington, DC, USA, Protocol for Isolation of Large RNA and Small RNA Fractions). 

To synthesise cDNA from mRNA, 0.34 μg of mRNA was used. cDNA synthesis was performed using the ImProm-II Reverse Transcription System (Promega, USA) and random hexamers, according to the manufacturer’s instructions. 

Before cDNA synthesis from miRNA, a small RNA fraction obtained by extraction was polyadenylated using the Poly(A)tailing kit (Life Technologies, USA), and cDNA was subsequently synthesised from 0.15 μg of the polyadenylated template using the ImProm-II™ Reverse Transcription kit (Promega, USA) and employing a primer with a universal tag [48] to extend the miRNA sequence.

To analyse mRNA and miRNA expression, a QuantiTect SYBR Green PCR Kit and a miScript SYBR green PCR kit (Qiagen, Hilden, Germany) were used, respectively. Quantification was performed using the StepOnePlus™ Real-Time PCR System (Applied Biosystems, Waltham, MA, USA). We used subsequent real-time PCR conditions: activation of hot start polymerase at 95 °C for 15 min, followed by 40 cycles at 94 °C for 15 s, 49–62 °C for 30 s (depending on a particular gene, see Appendix A) and extension at 72 °C for 30 s. The specificity of the PCR products was validated by melting curve analysis. Ribosomal protein *s17* was used for gene expression normalisation. The primers used in PCR reactions are provided in Appendix A.

## 5. Conclusions

In conclusion, the present study revealed new target genes of miR-34a-5p with key roles in the circadian oscillator functioning– *per2*, *bmal1* and *cry1*. In the case of *per2* and *bmal1*, miR-34a-5p downregulated gene expression, and this effect was most likely mediated via the 3′UTR region. *cry1* mRNA expression was upregulated after miR-34a administration, and this influence is more likely to be mediated by an alternative way of miRNA functioning.

Interestingly, exposure to RF-EMF potentiates the capacity of miR-34a to induce the expression of clock genes with oncogenic capacity—*cry1* and *cry2*. Moreover, the functioning of the miR-34a inhibitor was weakened when cells were exposed to RF-EMF, and consequently, the increase in tumour-suppressive *per2* mRNA expression induced by the miR-34a inhibitor was less pronounced under RF-EMF conditions compared to the control. Similarly, miR-34a inhibited the expression of anti-apoptotic protein *survivin* was diminished when cells were exposed to RF-EMF. RF-EMF have been classified by the WHO as possibly carcinogenic for humans. Our data are in line with this conclusion, as the results indicate that RF-EMF can shift the influence of miR-34a in cancer progression manifested by analysed target genes from typical tumour-suppressive to neutral or slightly oncogenic.

## Figures and Tables

**Figure 1 ijms-23-13210-f001:**
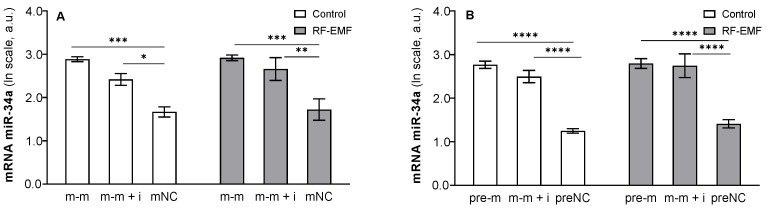
Levels of miR-34a-5p in the DLD1 cells. Cells were transfected with (**A**) miR-34a-5p (m-m), (**B**) pre-miR-34a (pre-m), mimic of mature miR-34a together with the miR-34a inhibitor (m-m + i) or corresponding negative control of mature and pre-miRNA (mNC a preNC, respectively). Data are presented as the mean ± SEM (n = 4) in logarithmic scale. a.u.—arbitrary units. Two-way ANOVA, followed by Tukey’s post hoc test, * *p* < 0.05; ** *p* < 0.01; *** *p* < 0.001; **** *p* < 0.001.

**Figure 2 ijms-23-13210-f002:**
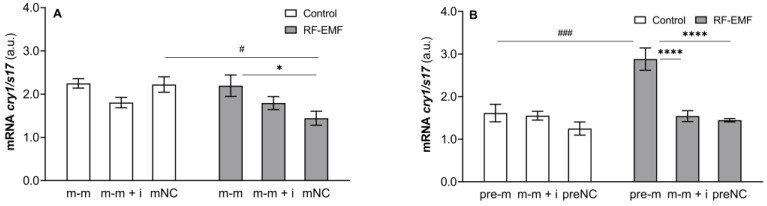
Effect of miR-34a administration on *cry1* mRNA expression in DLD1 cells under control and RF-EMF conditions. Cells were transfected with (**A**) miR-34a-5p (m-m), (**B**) pre-miR-34a (pre-m), mimic of mature miR-34a together with the miR-34a inhibitor (m-m + i) or corresponding negative control of mature and pre-miRNA (mNC and preNC, respectively). The data were relativised to the corresponding negative control measured under control conditions and presented as the mean ± SEM (n = 4). a.u.—arbitrary units. Two-way ANOVA, Tukey’s post hoc test, * *p* < 0.05, **** *p* < 0.0001—comparison between groups transfected with different oligos within the control or RF-EMF treatment. Two-way ANOVA, Šídák’s multiple comparisons test, # *p* < 0.05, ### *p* < 0.001—comparison between the control and RF-EMF groups.

**Figure 3 ijms-23-13210-f003:**
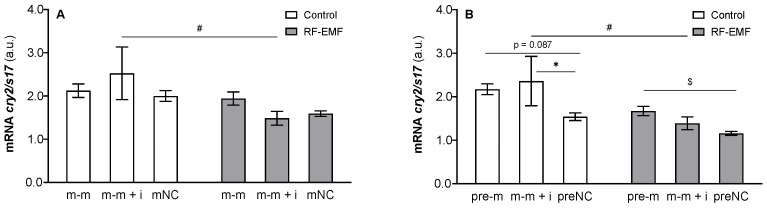
Effect of miR-34a administration on *cry2* mRNA expression in DLD1 cells under control and RF-EMF conditions. Cells were transfected with (**A**) miR-34a-5p (m-m), (**B**) pre-miR-34a (pre-m), (**A**,**B**) mimic of mature miR-34a together with the miR-34a inhibitor (m-m + i) or corresponding negative control of mature and pre-miRNA (mNC and preNC, respectively). The data were relativised to the corresponding negative control measured under control conditions and presented as the mean ± SEM (n = 4). a.u.—arbitrary units. Two-way ANOVA, Tukey’s posthoc test, * *p* < 0.05, comparison between groups transfected with different oligos within the control or RF-EMF treatment; One-way ANOVA, Tukey’s post hoc test, $ *p* < 0.05, comparison within the control or RF-EMF treatment. Two-way ANOVA, Šídák’s multiple comparisons test, # *p* < 0.05 comparison between the control and RF-EMF groups. For a more detailed statistical analysis, please see the results.

**Figure 4 ijms-23-13210-f004:**
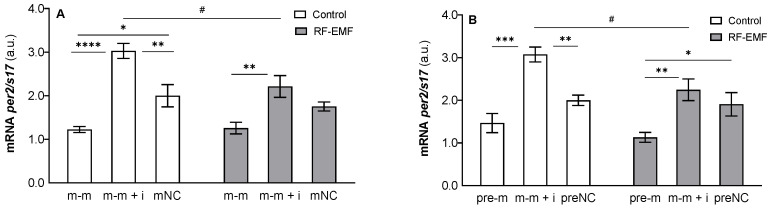
Effect of miR-34a administration on *per2* mRNA expression in DLD1 cells cultured in Wi-Fi-free conditions (white columns) or exposed to 24 h lasting RF-EMF exposure (grey columns). Cells were transfected with (**A**) miR-34a-5p (m-m), (**B**) pre-miR-34a (pre-m), mimic of mature miR-34a together with the miR-34a inhibitor (m-m + i) or corresponding negative control of mature and pre-miRNA (mNC and preNC, respectively). The data were relativised to the corresponding negative control measured under control conditions and presented as the mean ± SEM (n = 4). a.u.—arbitrary units. Two-way ANOVA, Tukey’s post hoc test, * *p* < 0.05, ** *p* < 0.01, *** *p* < 0.001, **** *p* < 0.0001—comparison between groups transfected with different oligos within the control or RF-EMF treatment. Two-way ANOVA, Šídák’s multiple comparisons test, # *p* < 0.05 comparison between the control and RF-EMF groups.

**Figure 5 ijms-23-13210-f005:**
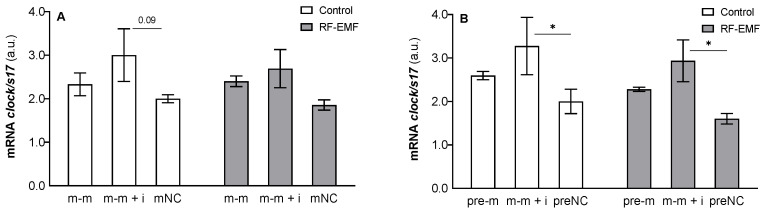
Effect of miR-34a administration on *clock* mRNA expression in DLD1 cells under control and RF-EMF conditions. Cells were transfected with (**A**) miR-34a-5p (m-m), (**B**) pre-miR-34a (pre-m), mimic of mature miR-34a together with the miR-34a inhibitor (m-m + i) or corresponding negative control of mature and pre-miRNA (mNC and preNC, respectively). The data were relativised to the corresponding negative control measured under control conditionspresented as the mean ± SEM (n = 4). a.u.—arbitrary units. Two-way ANOVA, Tukey’s post hoc test, * *p* < 0.05.

**Figure 6 ijms-23-13210-f006:**
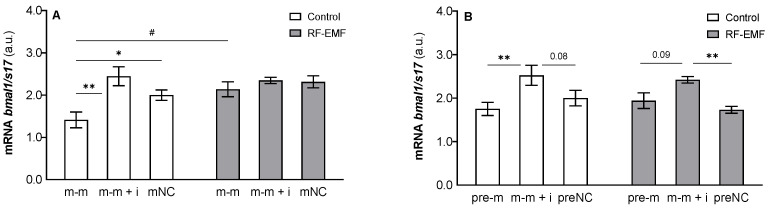
Effect of miR-34a administration on *bmal1* mRNA expression in DLD1 cells under control and RF-EMF conditions. Cells were transfected with (**A**) miR-34a-5p (m-m), (**B**) pre-miR-34a (pre-m), mimic of mature miR-34a together with the miR-34a inhibitor (m-m + i) or corresponding negative control of mature and pre-miRNA (mNC and preNC, respectively). The data were relativised to the corresponding negative control measured under control conditions and presented as the mean ± SEM (n = 4). a.u.—arbitrary units. Two-way ANOVA, Tukey’s post hoc test, * *p* < 0.05, ** *p* < 0.01, comparison between groups transfected with different oligos within the control or RF-EMF treatment. Two-way ANOVA, Šídák’s multiple comparisons test, # *p* < 0.05 comparison between control and RF-EMF groups.

**Figure 7 ijms-23-13210-f007:**
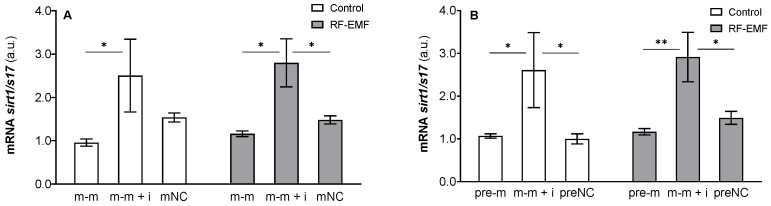
Effect of miR-34a administration on *sirt1* (*sirtuin 1*) mRNA expression in DLD1 cells under control and RF-EMF conditions. Cells were transfected with (**A**) miR-34a-5p (m-m), (**B**) pre-miR-34a (pre-m), mimic of mature miR-34a together with the miR-34a inhibitor (m-m + i) or corresponding negative control of mature and pre-miRNA (mNC and preNC, respectively). The data were relativised to the corresponding negative control measured under control conditions and presented as the mean ± SEM (n = 4). a.u.—arbitrary units. Two-way ANOVA, Tukey’s post hoc test, * *p* < 0.05, ** *p* < 0.01.

**Figure 8 ijms-23-13210-f008:**
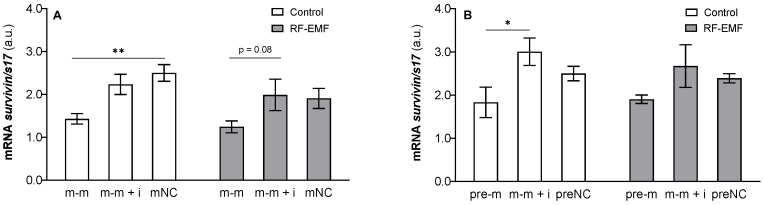
Effect of miR-34a administration on *survivin* mRNA expression in DLD1 cells under control and RF-EMF conditions. Cells were transfected with (**A**) miR-34a-5p (m-m), (**B**) pre-miR-34a (pre-m), mimic of mature miR-34a together with the miR-34a inhibitor (m-m + i) or corresponding negative control of mature and pre-miRNA (mNC and preNC, respectively). The data were relativised to the corresponding negative control measured under control conditions and presented as the mean ± SEM (n = 4). a.u.—arbitrary units. Two-way ANOVA, Tukey’s post hoc test, * *p* < 0.05, ** *p* < 0.01.

**Figure 9 ijms-23-13210-f009:**
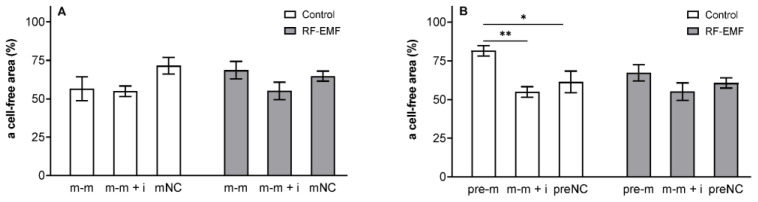
Effect of miR-34a administration on wound healing in DLD1 cells under control and RF-EMF conditions. Cells were transfected with (**A**) miR-34a-5p (m-m), (**B**) pre-miR-34a (pre-m), mimic of mature miR-34a together with the miR-34a inhibitor (m-m + i) or corresponding negative control of mature and pre-miRNA (mNC and preNC, respectively). Data obtained 24 h after wound generation were relativised to the corresponding cell-free area of the wound at time 0 h. Data are presented as the mean ± SEM (n = 4). Two-way ANOVA, Tukey’s post hoc test, * *p* < 0.05, ** *p* < 0.01.

**Figure 10 ijms-23-13210-f010:**
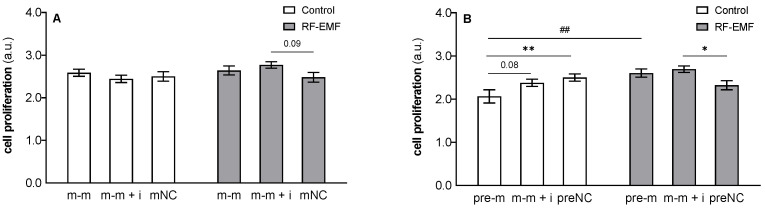
Effect of miR-34a administration on the proliferation of DLD1 cells under control and RF-EMF conditions. Cells were transfected with (**A**) miR-34a-5p (m-m), (**B**) pre-miR-34a (pre-m), mimic of mature miR-34a together with the miR-34a inhibitor (m-m + i) or corresponding negative control of mature and pre-miRNA (mNC and preNC, respectively) and incubated for 48 h; then, cell proliferation was assessed by a colorimetric MTS assay. Data are presented as the mean ± SEM (n = 4). Two-way ANOVA, Tukey’s post hoc test, * *p* < 0.05, ** *p* < 0.01, comparison between groups transfected with different oligos within control or RF-EMF treatment. Two-way ANOVA, Šídák’s multiple comparisons test, ## *p* < 0.01 comparison between control and RF-EMF groups.

## Data Availability

All relevant data within the document will be made available at Figshare.

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
