# Peer review of "2.4 GHz Electromagnetic Field Influences the Response of the Circadian Oscillator in the Colorectal Cancer Cell Line DLD1 to miR-34a-Mediated Regulation"

_ijms, 2022, doi:10.3390/ijms232113210_

Round 1

Reviewer 1 Report

Review on The 2.4 GHz electromagnetic field influences the response of the circadian oscillator in the colorectal cancer cell line DLD1 to miR-34a-mediated regulation

I have completed my review on manuscript ijms-1976477, entitled, The 2.4 GHz electromagnetic field influences the response of the circadian oscillator in the colorectal cancer cell line DLD1 to miR-34a-mediated regulation.” The radiofrequency electromagnetic field (RF-EMF) has pleiotropic effects on biological processes, including circadian rhythms. The proposed study sought to determine whether 24-hour exposure to 2.4 GHz RF-EMF influences miR-34a-induced changes in clock gene expression, migration, and proliferation in the colorectal cancer cell line DLD1. The authors conclude that RF-EMF strongly influenced regulation of the peripheral circadian oscillator in DLD1 cells via the tumor suppressor miR-34a.

The merit of this manuscript

Overall, the study was beneficial and resulted new findings. I believe this paper is worthy of publication in IJMS, but I have some questions and suggestions for the authors that must be addressed first.

Comments for authors

Comment 1: For new readers, the authors' literature review and background may not be sufficient to understand the significance of this study. It's also unclear how the electromagnetic field affects the cells in the manuscript. I recommend that authors include the suggested article in the introduction section to strengthen the background and mechanisms by which the EM field interacts and affect biological systems.

Article: Microwave Radiation and the Brain: Mechanisms, Current Status, and Future Prospects. International Journal of Molecular Sciences vol. 23 (2022). [https://doi.org/10.3390/ijms23169288].

Comment 2: Please explain how cells are exposed to a 2.4 GHz electromagnetic field in the material and method. If the authors include some sort of schematic, it will be more useful and easier to read/understand the manuscript. In the material and method section, each experiment has a distinct name for ease of reading.

Comment 3: What type of electromagnetic field was used for the exposure, pulsed or continuous? What about the thermal effect? When the authors indicate a 24-hour exposure time, I am concerned about temperature rise.

Comment 4: I recommend authors to include some literature on clock gene processes in various cancer cells in the introduction section.

Comment 5: Please specify in the results section how long the cells were cultured for miR-34a transfer. Is there any confirmation that miR-34a completely transforms cells?

Comment 6: There are typos and inaccuracies in the paper. I strongly recommend authors to read precisely and correct the grammatical errors.

Comment 7: In line 27, “peripheral tissues that facilitates proper synchronization” replace with “peripheral tissues that facilitate proper synchronization.”

Comment 8: When the short form of an electromagnetic field (EMF) has already been defined, avoid reusing it. Also, define the short form when it appears for the first time in the text. Authors are advised to preciously check in lines 71, 77, 402, 417, and 588.

Comment 9: In line 94, “four types of cancer (breast, lung and oesophageal) included in the analysis of” replace with “four types of cancer (breast, lung, and oesophageal) were included in the analysis.”

Comment 10: In line 431, “components” replace with “components.”

Comment 11: The conclusion should be written independently of the discussion section.

Author Response

Dear reviewer,

Thank you very much for your valuable time and opinion, we appreciate it very much. MS was edited by professional editing agency before submission (please, see attached certificate) and all authors carefully checked MS for typos and inaccuracies in the text. We hope that formal quality of text is sufficient now.

All your suggestions and comments were implemented into the text, please see detailed response bellow.

Comments for authors

Comment 1: For new readers, the authors' literature review and background may not be sufficient to understand the significance of this study. It's also unclear how the electromagnetic field affects the cells in the manuscript. I recommend that authors include the suggested article in the introduction section to strengthen the background and mechanisms by which the EM field interacts and affect biological systems.

Article: Microwave Radiation and the Brain: Mechanisms, Current Status, and Future Prospects. International Journal of Molecular Sciences vol. 23 (2022). [https://doi.org/10.3390/ijms23169288].

- thank you for the advice, we agree that suggested reference explains mechanism by which EMF can influence biological systems very well and therefore it was incorporated into MS. A new paragraph explaining mechanism of EMF on living organisms was incorporated into the chapter “Introduction”.

Comment 2: Please explain how cells are exposed to a 2.4 GHz electromagnetic field in the material and method. If the authors include some sort of schematic, it will be more useful and easier to read/understand the manuscript. In the material and method section, each experiment has a distinct name for ease of reading.

- thank you for the comment, a new scheme showing experimental set up was included into MS as supplementary Figure 1

- a distinct name was attributed to each experiment. These names are used also in scheme with experimental set up and chapter “Results”.

Comment 3: What type of electromagnetic field was used for the exposure, pulsed or continuous? What about the thermal effect? When the authors indicate a 24-hour exposure time, I am concerned about temperature rise.

- to investigated effects of 2.4 GHz EMF pulsed field with very short pulse duration (2.76 ms). Very short pulse duration is implicated also by RMS value that was only 0.12 W/m2 (6.6 V/m). EMF definition is provided in the 2nd and 3rd paragraph of the chapter “Materials and Methods”. Strengths of field used in present experiment was too low to cause a heating of cells cultured in temperature controlled incubator.

- to perform experiment a high quality new biological Celculture® Incubator CCL-050B-8 (Esco medical, Denmark) powered with a sophisticated firmware and robust hardware components to control temperature with sensor inside chamber was used. This incubator is able to control temperature in range from +7 to +60 oC, with temperature accuracy <± 0.1 oC.

We did not detect any rise in temperature during experiment. Therefore, we suppose that presence of Wi-Fi router did not influence temperature in the incubator in such a way, that incubator was not able to keep constant 37 oC.

Information about incubator used in the study was included into MS, chapter “Materials and Methods”.

Comment 4: I recommend authors to include some literature on clock gene processes in various cancer cells in the introduction section.

- thank you for the comment, section about clock gene expression was incorporated into the chapter “Introduction”

Comment 5: Please specify in the results section how long the cells were cultured for miR-34a transfer. Is there any confirmation that miR-34a completely transforms cells?

- we apologise for unclear description of experimental design. Oligos were added into medium immediately after trypsinisation and they were not washed away during the experiment. Therefore, miR-34a mimic was present in the medium during the whole experiment. This information is now better depicted in the Figure S1 that was added into revised version of MS.

- information was also provided in the results section in the first paragraph of text attributed to each particular experiment (qPCR, Scratch assay and MTS assay)

- efficiency of transformation is shown in figure 1, where logarithmic scale had to be used to show increase in the intracellular levels of miR-34a-5p in cells transfected with mature or premature form of miR-34a implicating very high efficiency of transformation.

Comment 6: There are typos and inaccuracies in the paper. I strongly recommend authors to read precisely and correct the grammatical errors.

- thank you for the advice, MS was carefully revised by all authors

Comment 7: In line 27, “peripheral tissues that facilitates proper synchronization” replace with “peripheral tissues that facilitate proper synchronization.”

- thank you very much, sentence was corrected

Comment 8: When the short form of an electromagnetic field (EMF) has already been defined, avoid reusing it. Also, define the short form when it appears for the first time in the text. Authors are advised to preciously check in lines 71, 77, 402, 417, and 588.

- we apologise for un-consistent use of abbreviation EMF. We omitted this abbreviation in the whole MS and use only RF EMF (for radio-frequency electromagnetic field). We replaced EMF by RF-EMF where it was appropriate.

Comment 9: In line 94, “four types of cancer (breast, lung and oesophageal) included in the analysis of” replace with “four types of cancer (breast, lung, and oesophageal) were included in the analysis.”

- thank you very much, sentence was corrected

Comment 10: In line 431, “components” replace with “components.”

- we omitted word “component” in the manuscript

Comment 11: The conclusion should be written independently of the discussion section.

- thank you for the comment, in revised form of MS the Conclusions are written independently form the chapter “Discussion”

Reviewer 2 Report

The manuscript was written well. It can be accepted.

Author Response

Dear reviewer,

we are very grateful for your time, opinion and greatly pleased by your evaluation.

Reviewer 3 Report

 I was really interested with the scope of the paper=

However I was reading the paper and asking what were the exposure conditions. Plese reconstruct the manuscript in normal order - Subjectq Introduction with the existing knowledge, Materials and methodsq Results

Please explain the reason for the choice of RF-EMF. Make the clarification for MF and EMF results and potential effects

Author Response

Dear reviewer,

Thank you very much for your valuable time and opinion, we appreciate it very much. We hope that changes that were made in MS according to your advice and suggestions are satisfying. Please see detailed response bellow.

I was really interested with the scope of the paper=

However I was reading the paper and asking what were the exposure conditions. Plese reconstruct the manuscript in normal order - Subjectq Introduction with the existing knowledge, Materials and methodsq Results

- thank you for the suggestion. Chapters of manuscript were re-arranged and section “Materials and methods” follows section “Introduction” in revised form of manuscript.

Please explain the reason for the choice of RF-EMF.

- thank you for the suggestion. We study effect of RF-EMF because its levels are constantly increasing in urban areas and reports about direct non-thermal effects of RF-EMF are accumulating. This reasoning is more emphasized in revised version of MS, please, see 4th and 5th paragraph of section “Introduction”.

Make the clarification for MF and EMF results and potential effects

- thank you for the comment. MF measured in our study at the cell layer was 0.3 µT, which is far bellow effective levels of MF used in studies focused on effect of magnetic field on living organisms, e.g.:

Mustafa E, Makinistian L, Luukkonen J, Juutilainen J, Naarala J. Do 50/60 Hz magnetic fields influence oxidative or DNA damage responses in human SH-SY5Y neuroblastoma cells? Int J Radiat Biol. 2022; 98(10):1581-1591.

Alkis ME, Akdag MZ, Kandemir SI. Influence of extremely low-frequency magnetic field on chemotherapy and electrochemotherapy efficacy in human Caco-2 colon cancer cells. Electromagn Biol Med. 2022;41(2):177-183.

Martiñón-Gutiérrez G, Luna-Castro M, Hernández-Muñoz R. Role of insulin/glucagon ratio and cell redox state in the hyperglycaemia induced by exposure to a 60-Hz magnetic field in rats. Sci Rep. 2021;11(1):11666.

This low MF is normally ubiquitous because of using of electric devices. Moreover, it was present in both, the experimental and control cells. Therefore, we suppose that under conditions of this study it did not cause differences between control and RF-EMF treated groups. Information that control as well as experimental groups were exposed to the same very low MF was included in the 3rd paragraph of section “Material and Methods”.

Round 2

Reviewer 1 Report

I have completed reviewing the changes made by the authors. I appreciate that the authors addressed my suggestions and concerns in this revised version. The paper is now worthy of publication in IJMS.